# 1 A low-cost, autonomous system for distributed snow depth

## 2 measurements on sea ice

- 3 Ian A. Raphael, Donald K. Perovich, Christopher M. Polashenski, Robert L. Hawley<sup>3</sup>
- <sup>1</sup>Thayer School of Engineering, Dartmouth College, Hanover, NH, USA
- <sup>2</sup>Cold Regions Research and Engineering Laboratory, US Army Corps of Engineers, Hanover, NH, USA
- <sup>3</sup>Department of Earth Sciences, Dartmouth College, Hanover, NH, USA
- Correspondence to: Ian A. Raphael (ian.a.raphael.th@dartmouth.edu)
- **Abstract.** Snow is a critical component of the Arctic sea ice system. With its low thermal conductivity and high
- albedo, snow moderates energy transfer between the atmosphere and ocean during both winter and summer, thereby
- playing a significant role in determining the magnitude, timing, and variability of sea ice growth and melt. The depth
- of snow on Arctic sea ice is highly variable in space and time, and accurate measurements of snow depth and
- variability are central to improving our basic understanding, model representation, and remote sensing observations
- of the Arctic system. Our ability to collect those measurements has hitherto been limited by the high cost and large
- size of existing autonomous snow measurement systems. We designed a new system called SnoTATOS (the Snow
- 15 Thickness and Temperature Observation System) to address this gap. SnoTATOS is a radio-networked, distributed
- snow depth observation system that is 95% less expensive and 93% lighter than existing systems. In this manuscript,
- we describe the technical specifications of the system and present results from a case study deployment of four
- 18 SnoTATOS networks (each with ten observing nodes) in the Lincoln Sea between April 2024 and January 2025.
- 19 The study demonstrates SnoTATOS' utility in collecting distributed, in situ snow depth, accumulation, and surface
- melt data. While surface melt varied within each network by up to 38%, mean surface melt between networks varied
- by only up to 9%. Similarly, whereas initial snow depth varied by up to 42% within each network, a comparison of
- mean initial snow depth between networks showed a maximum difference of only 26%. This indicates that floe-
- scale measurements made using SnoTATOS provide more representative data for regional intercomparisons than
- 24 existing single station systems. We conclude by recommending further research to determine the optimal number
- and arrangement of autonomous stations needed to capture the variability of snow depth on Arctic sea ice.

#### 1 Introduction

- September Arctic sea ice area has diminished by ~50% since satellite observations began in 1979 (Meier et al.,
- 2023; Onarheim et al., 2018; Peng and Meier, 2018). The remainder is predominantly thin first- and second-year ice
- (FYI, SYI) (Kwok, 2018). The Arctic Ocean may experience ice-free summers within the next decade (Jahn et al.,
- 2024). The thinning and loss of Arctic sea ice has increased Arctic coastal erosion (Barnhart et al., 2014; Eicken and
- Mahoney, 2015), diminished habitat (Laidre et al., 2015; Post et al., 2013), impeded hunting, fishing, and
- transportation over sea ice, and created new opportunities and uncertainties for shipping, tourism, military activity,
- and geopolitical conflict in the Arctic (Backus, 2012; Bystrowska, 2019; Carman, 2002). Understanding the Arctic
- ice pack is more important than ever. At the same time, rapidly changing conditions, in addition to baseline spatial

and temporal variability, present considerable challenges for our efforts to observe, understand, and predict changes in this environment.

Our fundamental understanding and model representations of the Arctic sea ice system are limited by the spatial and temporal resolution, consistency, coverage, representativeness, and scalability of available snow observations. Sturm et al. (2002) found early evidence that variability in snow cover properties, including snow depth, significantly impacts heat flow, surface temperatures, ice growth, and even marine mammals. Zampieri et al. (2024) and Clemens-Sewall et al. (2024b) both found that neglecting sub-meter to meter-scale snow depth variability results in a 10% underestimation of modeled conductive heat flux through the Arctic sea ice cover during winter, yielding a directly proportional underestimation of ice growth. Itkin and Liston (2024) identified meter to decimeter scale snow variability as a key control on area-average heat flux, and thus ice growth. Measurements of temperature profiles through the snow and ice are commonplace on observational campaigns. Augmenting these temperature measurements with precise, distributed snow depth observations enables investigators to determine the role of snow in controlling heat transfer at the local scale.

Snow also influences the timing of melt onset (Holland et al., 2021), and the formation and distribution of melt ponds (Polashenski et al., 2012), both of which impact the magnitude and spatial variability of sea ice melt. Clemens-Sewall et al. (2024b) and Holland et al. (2021) conclude that more observations of the spatial heterogeneity of snow depth are needed to improve model representations of sea ice conditions.

Further, snow accumulation can impact the validity of other measurements of the Arctic system. For example, an air temperature sensor initially installed at two meters effectively becomes a 1.5-meter air temperature sensor after 0.5 m of snow accumulation, with a similar result occurring due to surface melt. Accurate snow accumulation estimates are thus useful for interpretation of other datasets, especially those at the surface boundary layer.

Gerland et al. (2019) identified the sparsity of in situ measurements of snow depth as an essential gap in our understanding of Arctic sea ice, and in a review of snow in the contemporary sea ice system, Webster et al. (2018) stated that "Major questions remain ... as to the exact role of snow, how it varies regionally and seasonally, how snow conditions on sea ice are changing and what effects these changes have on the atmosphere—sea ice—ocean interactions," and that, "first and foremost, our limited understanding stems from the complexity of the snow—sea ice systems and the scarcity of observations." In short, we need high-spatial-resolution observations of snow depth to constrain spatial variability, validate remote sensing observations, advance model physics, and maintain an observational record of snow depth in the Arctic.

Remote sensing observations give broad and consistent geographical coverage, but do not afford the necessary spatial resolution or measurement precision (Meier and Markus, 2015; Webster et al., 2018). Crewed, in situ drift and station experiments – e.g., the 1997–1998 SHEBA expedition (Perovich et al., 1999, 2003; Sturm et al., 2002), the 2015 N-ICE experiment (Granskog et al., 2018; Merkouriadi et al., 2017; Rösel et al., 2018), the 2019–2020 MOSAiC expedition (Itkin et al., 2023; Nicolaus et al., 2022; Raphael et al., 2024), and the long-running Russian drifting ice station program (Colony et al., 1998; National Snow and Ice Data Center, 2004) – are important, but only partial, solutions. They provide opportunities to densely sample sea ice and snow conditions,

usually alongside a rich suite of atmosphere, ocean, and contextual information. However, each expedition offers only a snapshot in space and time.

Autonomous in situ instruments can provide wide spatial coverage and high temporal resolution, and several autonomous systems exist that offer precise, in situ measurements with selectable sampling frequency and up to 1–2 year endurance (Liao et al., 2019; Nicolaus et al., 2021; Planck et al., 2019). These systems are regularly deployed in the Arctic, but are expensive, heavy, and difficult to transport to and in the field (Table 1). This has historically limited their use to one to two instruments installed per floe, and few (<10) per region, the rare exceptions being major expeditions like N-ICE (Itkin et al., 2017; Nicolaus et al., 2021) and MOSAiC (Nicolaus et al., 2022; Rabe et al., 2024). Even on such major campaigns, relatively few units have been deployed on a single floe. These limited point measurements are usually taken as representative local snow depths. However, snow depth on Arctic sea ice can vary by two orders of magnitude over decimeter to kilometer length-scales due to topographical features, surface conditions of the underlying ice and snow, and ice age (and resultant accumulation time), among other factors (Clemens-Sewall et al., 2024a; Iacozza and Barber, 1999). A point measurement is unlikely to capture the mean (and, by definition, cannot capture the variance) of snow depth in complex local snow fields.

We need a new snow sensing technology that will improve the spatial density of Arctic snow depth measurements. The system must be inexpensive, easy to transport, use, and install, and have similar measurement precision and endurance to existing systems. We have designed, built, tested, and deployed the Snow Thickness and Temperature Observation System (SnoTATOS) to meet this need (Table 1). SnoTATOS is an autonomous, radionetworked, distributed snow depth measurement system that will accurately observe the mean and variance of snow depth on Arctic sea ice at meter to regional spatial scales. Throughout the design process, we focused on affordability; ease of manufacturing, transport, use, and deployment; and matching or exceeding existing measurement standards. Our ultimate goal is to reduce or eliminate barriers to deploying the system in large numbers across the Arctic. The SnoTATOS system is also a valuable tool for distributed observations of terrestrial snow, such as in alpine, tundra, and glacier environments.

In this manuscript, we describe the characteristics of the SnoTATOS system, share bench-testing performance evaluations, and present results from SnoTATOS prototype networks deployed in the Lincoln Sea in May 2024.

**Table 1:** Specifications of several polar snow depth measurement systems (all specifications are per unit/station)

| System                | Approximate cost | Weight | Size                            | Time to deploy | Endurance    | Measurement precision <sup>a</sup> |
|-----------------------|------------------|--------|---------------------------------|----------------|--------------|------------------------------------|
| MetOcean<br>Snow Buoy | \$9,400 USD      | 40 kg  | 2.55 m x 1<br>m x 1 m           | 30–40<br>min   | 12–18 months | ± 1 mm                             |
| SAMS<br>SIMBA buoy    | \$10,000 USD     | 25 kg  | ~0.55 m x<br>0.30 m x<br>0.20 m | 20–30<br>min   | > 12 months  | ± 2 cm                             |

| SIMB3    | \$18,000 USD           | 36 kg  | 4.87 m x<br>0.25 m x<br>0.11 m | 20–30<br>min | 24 months              | ± 1 mm |
|----------|------------------------|--------|--------------------------------|--------------|------------------------|--------|
| SnoTATOS | \$500 USD <sup>b</sup> | 1.8 kg | 2.44 m x<br>0.15 m x<br>0.1 m  | <10 min      | 4.5 years <sup>c</sup> | ± 1 mm |

<sup>a</sup>This value specifies the instrument's stated measurement precision, not the accuracy of the snow depth retrieval. The precision of the ultrasonic rangefinders is  $\pm 1$  mm, while accuracy depends on temperature compensation, ice/snow surface conditions, sensor icing, etc. The precision of digital temperature chain instruments (e.g., the SIMBA) is  $\pm 2$  cm (the separation between any two temperature sensors in the chain), while the accuracy depends on the thermal characteristics of the snow, ice, and atmosphere, which affect the feasibility of determining the interfaces between the three media.

<sup>b</sup>Cost of components only (not including manufacturing and assembly) is approximately \$200.

<sup>c</sup>This is a nominal endurance based on power consumption measurements in a laboratory setting. We expect the effective endurance to be reduced by low temperatures and any radio communication reattempts.

### 2 System description

#### 2.1 Overview of SnoTATOS

A standard SnoTATOS network consists of several autonomous snow measurement stations (hereafter called "nodes") linked to a central server by a LoRa radio network (Augustin et al., 2016) (Fig. 1). The number of nodes in a network is theoretically unlimited. Each node is equipped with an ultrasonic rangefinder (HRXL-MaxSonar-WR Datasheet, 2024) for monitoring the snow or ice surface position; additional sensors (e.g. temperature sensors) can be added with minimal engineering effort. The network is synchronized such that all nodes simultaneously collect samples and transmit their data back to the server at regular intervals, with random transmission jitter introduced to reduce packet collisions. The sampling frequency is programmable with a typical interval set at four hours. We initially designed the server to integrate into a SIMB3 ice mass balance buoy (Planck et al., 2019, p.201), thereby taking advantage of the SIMB3's existing Iridium telemetry. We have since redesigned the server to operate in a freestanding mode, either transmitting data using its own Iridium telemetry module or storing data locally on an SD card. In the following sections, we will describe the node and server electronics, physical characteristics, radio network, and operating software.

**Figure 1: Diagram of a SnoTATOS network**. SnoTATOS data is collected at each node in a distributed network, and transferred to the server via radio, either directly (in the hub-and-spoke network model) or via relay through peers (in the mesh network model). The server collects all SnoTATOS data and relays it to the SIMB3, which handles satellite telemetry to a land-side server.

### 2.2 Node overview and physical characteristics

A SnoTATOS node consists of a MaxBotix 7389-200 ultrasonic surface rangefinder; a microcontroller that manages sampling, datalogging, and radio communications with the server; a nickel-metal hydride (NiMH) battery powerbank; and ancillary electronics. Figure A1 shows a system block diagram. The electronics are housed in a watertight plastic enclosure (Fig. 2). The rangefinder is mounted directly to a sidewall of the enclosure (Fig. 2). The resulting sensing unit is 0.08 m x 0.19 m x 0.09 cm and weighs approximately 0.62 kg. The sensing unit is mounted on a 2.44 m x 0.038 m x 0.038 m (8 feet x 1.5 inches x 1.5 inches) wooden stake (Fig. 3). The long edges of the stake are filleted so that the stake fits snugly in a standard 5 cm (2 inch) diameter ice auger hole. The total weight of an individual node is approximately 1.80 kg, representing a 96% mass reduction compared to the MetOcean Snow Buoy. The stake length maximizes the range of observable snow depths while ensuring ease of transport to field sites by conforming to less-than-truckload (LTL) and passenger aircraft lower deck freight limitations, where freight often must be less than approximately 2.44 m long.

**Figure 2: SnoTATOS node sensing unit.** Panel (a) is a photograph of a SnoTATOS sensing unit, showing the ABS plastic enclosure and ultrasonic surface rangefinder mounted in the sidewall of the enclosure. Panel (b) is a top down photograph of the sensing unit with lid removed, showing the PCB, rangefinder wiring harness, and battery bank. Panel (c) shows an annotated digital model of a node PCB with key features identified.

The Maxbotix ultrasonic rangefinder detection cone has an approximately  $40^{\circ}$  aperture angle, so spurious detection of the mounting stake was a significant design concern. We conducted a series of experiments to determine the optimal sensor look-angle ( $\theta$ ) and standoff of the sensor from the mounting stake. We determined that a sensor standoff between 5–40 cm and  $5^{\circ} 

**Figure 3: Schematic diagram of a SnoTATOS node**. Snow depth  $(h_s)$  can be calculated by subtracting the range reading  $(R_s)$  from the rangefinder offset  $(Z_p)$ .

#### 2.3 Sensing unit electronics

Here, we summarize the selection of key components in the sensing unit and their notable features. The sensing unit is built around an ATmega4808 AVR microcontroller unit (MCU). The ATmega4808 is an 8-bit reduced instruction set computer (RISC) (Patterson, 1985) with 48 KB of program memory and 6 KB of RAM. The chip is equipped with an onboard 10-bit analog-to-digital converter (ADC). We added an external crystal oscillator which drives a one-second precision system clock, enabling an ultra-low-power standby mode with programmable, alarmed wakeups. In standby mode, unused peripheral devices are depowered and the MCU sleeps until woken, either by a programmed alarm or by an external interrupt on a general-purpose input/output (GPIO) pin. We selected the ATmega4808 for its low power consumption, affordability, and programming simplicity. The MCU has an operating input voltage range of 1.8-5.5 V, however, logic levels and GPIO output voltage are dependent on MCU input voltage. We added a low-quiescent-current ( $0.3~\mu$ A) buck-boost converter with a 1.8-5.5 V input voltage range and a fixed 3.3 V output. This achieves 3.3 V board logic and GPIO output voltage while maintaining flexibility in power supply voltage (Table 2).

We selected the HopeRF RFM95-915 LoRa module for radio communications. The module operates at 915 MHz with a maximum output power of 20 dBm. The 902–928 MHz frequency range is a license-free Industrial, Scientific, and Medical (ISM) radio band in the Americas (including the United States, Greenland, Canada, and South and Central America). The unit is directly exchangeable for the RFM95-868, which operates at 868 MHz, within the European ISM band (including the Russian Federation). These two options ensure system compliance for any Arctic deployment. Either option is suitable for deployments in international waters. The authors are not aware of any regulations restricting radio frequency use in Antarctica.

Most snow accumulation observation systems use one of several models of the Maxbotix ultrasonic rangefinder. Maxbotix offers many variations of their basic rangefinder, including snow-specific models. We chose to use their general-purpose model with the compact horn option (MB7389-200).

We use NiMH batteries for the power bank due to their improved cold-weather performance vs. alkaline batteries (Fetcenko et al., 2007) and less stringent shipping regulations compared to lithium-ion batteries. We used Tenergy Power D-cells, rated to 10,000 mAh per cell. Each node has a power bank of 4-cells, arranged in two parallel pairs of two cells in series. A NiMH battery has a functional voltage of  $\sim$ 1.2 V for most of its discharge life in normal conditions, yielding a nominal supply voltage  $V_{\rm n} = 2.4$  V and a nominal energy capacity of 24 Wh.

We designed a custom printed circuit board (PCB) to integrate all components (Fig. 2). The PCB is a two-layer board designed on a 1.6 mm FR-4 substrate. We designed a monopole PCB trace radio antenna adapted from a Texas Instruments design (Wallace, 2013).

Table 2: SnoTATOS electrical characteristics

|                           | Nominal input voltage | Input voltage operating range | Average power demand |
|---------------------------|-----------------------|-------------------------------|----------------------|
| Node                      | 2.4 V                 | 1.8-5.5 V                     | 610 μW               |
| Server (standalone)       | 9.0 V                 | 5.1-36.0 V                    | 15.54 mW             |
| Server (SIMB3 integrated) | 18 V                  | 3.4-36.0 V                    | 18.54 mW             |

### 2.4 Server electronics

The SnoTATOS server uses the same MCU, radio module, and antenna design as the node sensing units. However, the server is not equipped with sensors. In freestanding mode, the server is equipped with a RockBLOCK 9603 Iridium Short Burst Data (SBD) modem. The standalone server is equipped with a 9 V, 388 Wh NiMH power supply. A Pololu D24V5F5 buck converter steps the supply voltage down to 5 V to supply the RockBLOCK 9603 unit, and a Pololu D24V5F3 buck converter steps the supply voltage down to 3.3 V to supply the server MCU.

When integrated into the SIMB3 buoy, the server is designed to use the SIMB3's 18 V power supply. We used a Pololu D24V5F5 buck converter to step the 18 V SIMB3 supply down to 5 V to supply the server MCU. We integrated all components using a custom PCB similar to the node PCB.

#### 2.5 Software

#### 2.5.1 Node operations

The system software is written in C and C++, using the Arduino hardware abstraction layer (HAL) to interface with the MCU. The nodes follow the high-level logical flow shown in Fig. 4. When powered on, the node enters the Setup function, where it initializes the memory state, system clock, radio module, and rangefinder, and sets input/output pin states. The node then moves into the Loop function, where it will remain for its lifetime unless it is power-cycled. In Loop, the node first samples the rangefinder to obtain a snow depth reading at "wake-up" time. The node then checks its synchronization state. If it is not synced with the server (as is the case upon initial power-up), it will wait at this stage until it receives a synchronization broadcast message from the server. After synchronizing with the server, the node immediately sets an RTC alarm to wake after the appropriate elapsed time (the sampling interval). The node then reads its battery voltage, packs this and the rangefinder data into a buffer, and attempts to transmit the buffer to the server. If the transmission is successful and acknowledged by the server, the node depowers all unnecessary peripherals and enters a deep sleep state until triggered by the RTC alarm. However, if more than three *unsuccessful/unacknowledged* transmissions occur, the node returns to an unsynced state and remains awake until resyncing with the server. We implemented this failsafe to prevent network failure in case of clock drift or other errors resulting in network desynchronization over the course of the deployment.

**Figure 4: Node flow diagram.** The high-level logic of a SnoTATOS node equipped with only a snow surface rangefinder is shown. Additional sensors may be added, which would be read at the same stage as the surface rangefinder.

### 2.5.2 Radio communications

The radio network is implemented using LoRa, a long-range, low-power radio technology (Augustin et al., 2016). Nominal LoRa radio ranges are up to 10–20 km with clear line of sight. The RFM95 LoRa transceiver manages the physical layer of the Open Systems Interconnection (OSI) network model (Zimmermann, 1980), handling bitwise

data encoding, chirp spread spectrum (CSS) modulation, and physical transmission of the data. We used the open source RadioLib library (RadioLib - Arduino Reference, 2024) to implement the data link layer atop the physical layer; this handles data-packet to dataframe formatting and the digital interface between the MCU and the RFM95 module.

We developed software to implement the Network, Transport, Session, and Presentation layers of the OSI model. These handle data packet assembly, addressed packet transmission, packet receipt acknowledgment, failed transmission reattempts, packet transmission timeouts, and network collision handling. These are well established general concepts in computer networking, which we implemented in a lightweight C++ library for handling small-packet data transmission in an addressed, reliable network, with options for either hub-and-spoke or mesh network topologies. We gave particular attention to robust packet acknowledgement and secure server—node transactions, since this reduces network airtime for each node (by preventing unnecessary reattempts), in turn reducing potential node—node collisions. This results in a more reliable network, with less power expended on multiple transmission reattempts and unnecessary node waketime.

The system's standard network topology is the hub-and-spoke model, where individual nodes (the "spokes") communicate directly with the server (the "hub"). This network topology is simple to implement and is also typically the least power-intensive network model. In this topology, network sizes are limited by the 10–20 km nominal LoRa range. This range assumes line-of-sight between node and server. However, range and reliability may be reduced in complex terrain, such as in highly deformed sea ice where direct line-of-sight between the server and each node may not be possible since the sensing unit sits approximately 1.44 m above the ice surface (Figure 3), while ridges can reach a height of several meters (Duncan et al., 2018). We implemented an alternative mesh network topology to address this limitation, where data from out-of-range nodes can be relayed to the server through peers. We use a naive flooding protocol (Zahn et al., 2009) with acknowledged packet receipt. A detailed description of the node-side mesh network implementation is included in Appendix B.

### 2.5.3 Server operations

The server follows the high-level logical flow shown in Fig. 5. When powered on, the server enters the Setup function, where it initializes its memory state, system clock, radio module, SIMB3 communications (if integrated into SIMB3), and sets input/output pin states. The server then moves into the Loop function, where it will remain for its lifetime unless it is power-cycled. In Loop, the server first sets a "bedtime" alarm, which will trigger when the server wake-period ends and it is time for the server to enter standby mode. It then broadcasts a sync message to the network, and proceeds to loop through two stages until the bedtime alarm triggers.

In the first stage, the server checks to see if it has received a message from a node. If it has, it writes the node's data to the appropriate location in its memory buffer for later Iridium telemetry (standalone server) or transfer to SIMB3 (integrated server), then returns an ACK message to the originating node. In the standard huband-spoke topology, this is a unicast message directly to the originating node. A description of the server-side mesh network operations is included in the Appendix B.

If the server is operating in standlone mode, it continues to listen for data from nodes until it is time to sleep. When it is time to sleep, the server sets a wakeup alarm corresponding to the sampling interval, transmits its data via Iridium, and enters standby mode. If the server is integrated into a SIMB3, the server adds a second stage during its waketime loop. In the second stage, the server checks to see if the SIMB3 has requested the data from the server. If the SIMB3 has requested data, the server passes the buffer to the SIMB3, then resets the buffer to default values. The server continues checking these two conditions ("Received data from a node?" and "SIMB3 requested data?") until it is time to sleep, at which point it will set a wakeup alarm and enter standby mode. Despite the server checking the "SIMB3 requested data?" condition multiple times, the SIMB3 is expected to request data only once during a given sampling interval. However, due to communications protocols between the SIMB3 and the server, it is beneficial to respond to any hypothetical SIMB3 request as legitimate, even if the server responds with default buffer values.

Under normal conditions, all nodes are expected to have transmitted their data to the server before the server transmits data, either via Iridium or to the SIMB3. The server will not wait for all nodes to transmit before transmitting data; this prevents the server from becoming unresponsive if a node fails to transmit or is otherwise inoperable.

**Figure 5: Server flow diagram.** Panel (a.) shows the high-level logic flow for a standalone SnoTATOS server, and panel (b.) shows the high-level logic for a server integrated into a SIMB3 buoy.

### 2.6 SIMB3 integration

We used the I2C (Inter-Integrated Circuit) protocol to establish communications and data transfer between the server and the SIMB3. I2C is a serial communication protocol that allows a controller device (in this case, the SIMB3) to query packetized data from an addressed target device (the server). In addition to the standard I2C SDA (serial data) and SCL (serial clock) lines, we added a low-active chip select line (CS). The server and SIMB3 share a common ground line. When the SIMB3 is preparing to retrieve data from the server, it sets the CS line to ground (0 V) to notify the server. The server then prepares the data buffer for the SIMB3 and stands by until the SIMB3 retrieves the data through an I2C request or the transaction has timed out. The SIMB3 adds the retrieved data to its existing Iridium message and transmits it to a land-side server.

### 2.7 Bench tested power characteristics

We performed laboratory tests to estimate the power characteristics of the sensing unit using the shunt-resistor method and linear circuit analysis. By measuring the voltage drop,  $V_r$ , across a resistor with a known and low value, R, one can use Ohm's law ( $V_r = IR$ ) to determine the corresponding circuit current, I. With a known supply voltage,  $V_s$ , one can then use the power law (P = IV) to determine the circuit power demand, P. We used an oscilloscope to make time-resolved voltage measurements through all phases of the node's operating cycle, then converted these measurements to time-resolved power (Fig. A2).

We tested over a range of supply voltages that the node might typically experience, from  $V_s = 1.6$  V (below the buck/boost converter threshold voltage of 1.8 V) to  $V_s = 3.3$  V (above the nominal battery bank supply voltage,  $V_n = 2.4$  V). We determined that at  $V_s = V_n = 2.4$  V, the average circuit current across all phases of the typical 4-hour duty cycle is 254  $\mu$ A, and the average power demand is 610  $\mu$ W. With a 24 Wh power bank (two 10,000 mAh D-cell batteries), each node has an estimated endurance of ~1,639 days, or ~4.5 years (far longer than the lifetime of any sea ice on which it is likely to be installed). However, this does not account for battery efficiency losses due to cold temperatures, nor atypical conditions such as radio transmission retries.

We conducted similar power tests for the server, finding an average current draw of 1.03 mA at  $V_s = 18$  V, yielding an average power demand of 18.54 mW. This is approximately 30% of the SIMB3's power budget (Planck, 2021), yielding an estimated endurance of approximately 560 days, or slightly more than 1.5 years. Operating in standalone mode, the power supply can be reduced to  $V_s = 3.4$  V, increasing efficiency and reducing average power demand to approximately 2,500  $\mu$ W. This produces a nominal endurance of 4.4 years with a 96 Wh battery bank (eight 10,000 mAh D-cell batteries).

### 3 Case study, Lincoln Sea, April 2024–January 2025

We deployed four SnoTATOS networks in the Lincoln Sea in late April and early May, 2024, during the NASA ARCSIX project (McNamee, 2024) (Fig. 6). Each network consisted of ten nodes and a server integrated into a SIMB3 buoy. We deployed the networks in multiyear ice just before the onset of surface melt. We placed the nodes randomly between 25 and 200 m from each buoy, with clear line-of-sight to the buoy. We measured initial snow depth at each node, and ice thickness and snow depth at each SIMB3. As of 3 January, 2025, three networks (2024L, 2024O, and 2024R) were no longer reporting. The failure of 2024O is consistent with an I2C communications failure between the server and SIMB3 MCU. The steady attrition of nodes and their location in a shear band suggest that networks 2024L and 2024R were destroyed by ice dynamics. 2024P continues to report, with four nodes surviving; the rest were likely destroyed by ice dynamics. We will now describe the general results from these installations. We include data from network 2024O in summary visualizations for completeness, however, we do not consider these data in our analysis.

Figure 6: Drift tracks of four SnoTATOS networks deployed in the Lincoln Sea in April and May, 2024.

The mean conditions for all nodes at the time of installation of each of the four networks are given in Table 3. The time series of snow depth and surface melt for all nodes at each network is shown in Fig. 7. We observed between 0.05 and 0.10 m of snow accumulation at each network between installation in late April and late May. Surface melt in the region began in late May, after which snow depth decreased steadily at all nodes, reaching 0 m between 12 June and 8 July. On average, snow persisted longest at network P, which also had the deepest initial snow cover (Fig. 8). Ice surface melt then commenced, continuing until early August (Fig. 9).

Figure 7: Time series scatterplots of surface position at four SnoTATOS networks. Time series data of surface position is shown for each node at the four ARCSIX SnoTATOS networks. "Surface position" is the position of the surface sensed by the ultrasonic rangefinder (air—snow or air—ice interface) relative to the initial snow—ice interface (surface position 0). Each node initially demonstrates a positive surface position value, indicating a positive snow depth. Snow depth increases until around early June at all nodes. Snow melt then begins around mid-June, continuing at each node until the surface position reaches 0, indicating complete snow melt and the onset of ice surface melt. Ice surface melt continues until early August. From that point on, any positive change in surface position indicates new snow accumulation.

The results show substantial variability in initial snow depth, magnitude and timing of surface melt, and snow accumulation. Mean initial snow depths varied between networks by up to 26% (0.23 m at R vs. 0.31 m at L and P). Within the networks, initial snow depth variability ranged from 26% at network R to 42% at network L.

**Table 3: ARCSIX summary conditions** 

| Network<br>name | Duration | Initial ice<br>thickness<br>(m) | Mean initial<br>snow depth ±<br>standard<br>deviation (m) | Mean ice<br>surface<br>melt (m) | Mean<br>combined ice<br>equivalent<br>surface melt<br>(m) | Site description |
|-----------------|----------|---------------------------------|-----------------------------------------------------------|---------------------------------|-----------------------------------------------------------|------------------|
|-----------------|----------|---------------------------------|-----------------------------------------------------------|---------------------------------|-----------------------------------------------------------|------------------|

| 2024L | 29 April–1<br>November,<br>2024                 | 1.96 | $0.31 \pm 0.13$ | 0.23 ± 0.11 | $0.33 \pm 0.08$ | Level multiyear ice (MYI) floe. Potential hummocks which snow has filled, rendering a smooth surface.     |
|-------|-------------------------------------------------|------|-----------------|-------------|-----------------|-----------------------------------------------------------------------------------------------------------|
| 20240 | 5 May-1<br>June, 2024                           | 1.72 | $0.29 \pm 0.09$ | ~           | ~               | Large MYI or SYI<br>pan with relatively<br>level surface. May<br>have experienced<br>little surface melt. |
| 2024P | 6 May, 2024–<br>3 January,<br>2025 <sup>a</sup> | 2.16 | 0.31 ± 0.10     | 0.20 ± 0.06 | $0.31 \pm 0.05$ | Hummocky MYI floe in ridged area. Floe too thick to drill in some places (> 4 m).                         |
| 2024R | 4 May–25<br>November,<br>2024                   | 2.40 | $0.23 \pm 0.06$ | 0.23 ± 0.11 | $0.30 \pm 0.11$ | Hummocky MYI floe.                                                                                        |

<sup>356 &</sup>lt;sup>a</sup>Four nodes from network P were still reporting as of 3 January, 2025.

**Figure 8: Box-and-whisker time series of surface position at four SnoTATOS networks.** Each box-and-whisker shows the spatial distribution of the ten-day-average surface position for a given network. The lower and upper edge of each box show the first and third quartiles, the bar in the box shows the median, and the whiskers indicate the minimum and maximum non-outlier values. Outliers are shown as open blue circles, and are defined as more than 1.5 times the interquartile range lesser or greater than the first and third quartiles, respectively. The small, dotted markers and interpolated line show the spatial mean for each ten-day bin. The square, grey markers indicate the sample size (number of nodes) included in the distribution at each time step, with a separate Y-axis shown on the right of each pane.

We computed the ice equivalent snow melt (snow-ice equivalent; SIE) using Eq. 1:

$$H_{\rm sie} = \rho_{\rm s}/\rho_{\rm i} * H_{\rm snow} \,, \tag{1}$$

where  $\rho_i$  is the density of sea ice (0.9 g cm<sup>-3</sup>, Perovich et al., 2003),  $\rho_s$  is the density of snow (0.3 g cm<sup>-3</sup>, Sturm et al., 2002),  $H_{snow}$  is the observed snow melt, and  $H_{sie}$  is the SIE melt. We combined  $H_{sie}$  with the observed ice surface melt to determine the total ice equivalent surface melt for each station. Average ice-equivalent melt was 0.33 m at L, 0.31 m at P and 0.30 m at R, indicating very similar net surface melt across the region. Net ice-only surface melts were also quite similar with 0.23 m at L, 0.20 m at P and 0.23 m at R. The network with the deepest initial snow depth (P) also had the smallest ice melt, presumably because deeper snow increased albedo and physically

protected the ice, delaying surface melt onset (Fig. 9). Compared to variability between regions or years within the Arctic (e.g., Perovich (2014) or Planck (2022)), however, these variations in mean behavior are quite small.

A key note here is that variability in surface melt (both ice surface melt and combined equivalent melt) was relatively low *between* networks, the largest variability being a 13% difference in ice surface melt between R and P (R higher), and a 9% difference in combined equivalent melt between L and R (L higher). However, melt variability *within* networks was higher, at 31–46% for ice surface melt, and 15–38% for combined equivalent melt. This suggests that networks of this size (on the order of ten nodes) may be adequate for accurately capturing the local variability of surface melt. We note that the surface melt variability seen here was lower than on SHEBA and MOSAiC, where the maximum differences in observed surface ice melt were 55% and 71% (Perovich, 2002; Raphael, 2024). We recommend a more thorough evaluation of the number of stations required to capture surface melt variability.

Snow accumulation began soon after the conclusion of surface melt, in early to mid August. Network L saw 0.08 m snow accumulation by 16 October, then a decrease to 0.04 m snow depth by 26 October, when the network ceased reporting. The air temperature record suggests that the decrease was caused by wind removal rather than surface melt. Network R saw 0.14 m of new snow by 15 November, when it also ceased returning data. As of 3 January, 2025, network P has seen a mean snow accumulation of 0.39 m and a range of 0.12–0.80 m.

Figure 9: Box-and-whisker plot showing the distribution of ice surface melt onset and surface melt end dates. Ice surface melt onset is shown in orange, and surface melt end is shown in blue for the nodes within each network. Network 2024O is excluded since the network stopped reporting before surface melt onset. "All" shows the combined distribution of all active nodes in 2024R, 2024P, and 2024L.

Despite relatively small geographical separation, snow accumulation varied significantly between networks. We compare the snow accumulation at networks L, P, and R during the period from freezeup around early August, through 26 October, when network L failed. The networks were deployed within 113 km of each other, and by 26 October, networks L and P were still within 98 km of each other. Meanwhile, network R drifted to 306 km from network L, and 398 km from network P. During this period, 0.04 m of snow accumulated at network L, 0.25 m of snow accumulated at neftwork P, and 0.14 m of snow accumulated at network R. This indicates a roughly 84% difference in snow accumulation between networks L and P in that period, despite their relative proximity.

Further, the variability of snow accumulation within each network is evident in the widening box-and-whisker distributions in Fig. 8. This variability increases as accumulation continues through the winter at network R and, in particular, at network P. The attrition of nodes at network P during this period prompted us to consider whether the increase in the interquartile range (IQR) is an artifact of the declining sampling size or a real signal. Because the increase in IQR occurs primarily during a period when the sample size is constant (n = 4), we suggest that the increase in the IQR is a real signal that is amplified by the small sample size.

Finally, the range of snow depth on 26 October was approximately equal to the range at time of installation for network L, slightly higher at network R, and substantially higher at network P. This is potentially the result of both interannual as well as spatial variability (due to ice advection).

As many studies have confirmed, snow depth on sea ice is highly variable; this case study suggests that SnoTATOS can observe that variability, though the number of nodes needed to fully constrain it is unclear. In order to facilitate efficient use of resources and enable accurate, error-constrained data collection, we recommend further research into the number and arrangement of sampling points needed to measure the spatial and temporal variability of the snow cover on Arctic sea ice. Such a study should investigate the errors produced when using various sample sizes and patterns to estimate snow depth mean and variance at the floe scale, and, ultimately, identify the minimum number of stations typically needed to constrain these statistics. Sturm (2009) conducted a limited study by resampling snow depth transect data with consecutively decreasing sample sizes, however, this study was limited to three, one-dimensional transects, all collected in the same location on the same date. A similar, more extensive study should be undertaken by resampling data collected across multiple locations, instances, and ice types. We also suggest testing various spatial arrangements of the sample points (random, gridded, etc.).

#### 4 Conclusions

This work documents the development, testing, and a case study deployment of SnoTATOS, a new autonomous system for collecting distributed, in situ snow depth measurements on sea ice. Responding to community calls for the widespread snow depth measurements that are needed to understand the changing Arctic sea ice system, and recognizing the lack of suitable, affordable tools, we set out to create a low-cost, easy-to-use system to fill the gap. The resulting radio-networked snow depth measurement stations are only 5% of the cost and 7% of the weight of

existing systems, with identical measurement functionality. A case study deployment of four SnoTATOS networks in the Lincoln Sea in April 2024 1) validates the functionality of SnoTATOS, including the system's ease of transport, rapid installation, and collection of high-quality, in situ snow depth and surface melt measurements, 2) demonstrates the substantial spatial and temporal variability in snow accumulation and ice surface melt at the floe scale, and 3) suggests that even relatively small SnoTATOS networks (on the order of 10 nodes) are capable of capturing that variability. Based on the last finding, we recommend focused studies to determine the number and placement of autonomous sampling stations needed to accurately capture snow accumulation, depth, and surface melt variability.

Of the forty nodes installed in April 2024, four were still reporting by the beginning of January 2025. The character of the failures suggests most (26) failed by physical damage. High attrition rates resulting from ice dynamics and wildlife are a reality for autonomous instruments installed on Arctic sea ice. This, in addition to a need for more comprehensive observations of Arctic variability, is a strong motivation to transition towards the use of large, redundant networks of lightweight, inexpensive sensing stations, an approach also recommended by Lee et al. (2022) and Webster et al. (2022). In its current permutation, SnoTATOS can accommodate additional sensors such as barometric pressure or temperature sensors. We plan to build on this technology to create a modular "polar Internet of Things" sensing system capable of hosting plug-and-play sensors, making radio-networked distributed sensing more accessible for the polar regions. We anticipate that SnoTATOS will also prove useful for monitoring snow accumulation and ice surface melt in alpine, glacier, and tundra environments.

### Appendix A: sensing unit components and power test

**Figure A1: Schematic block diagram of SnoTATOS sensing unit electronics.** The figure shows the major electronics components of the SnoTATOS sensing unit. Blue blocks indicate external power and clock components for the MCU, which is shown in orange. Yellow blocks indicate I/O modules that the MCU interacts with for collecting and transmitting data.

Figure A2: Time-resolved power demand for the node and server during pre-deployment bench testing. Panel (a.) shows the power demand during the various stages of the duty cycle for a node with  $V_s = 2.4 \text{ V}$ . Panel (b.) shows the power demand during the various stages of the duty cycle for the server with  $V_s = 18 \text{ V}$ .

### Appendix B: mesh network implementation

The node-side logical flow for mesh network packet handling is shown in Fig. B1. During a data transmission attempt, a node will first attempt to unicast the message directly to the server. If an acknowledgment (ACK) is received, then the message has been transmitted successfully and the attempt ends. If an ACK is not received within a timeout period, the node then reattempts transmission, either repeating a unicast if the last ACK'd message was *not* a broadcast, or progressing directly to broadcast attempts if the node knows that the last message it successfully transmitted to the server was a broadcast message. If an ACK is not received within the allotted number of reattempts, or the timeout period expires, then the transmission attempt has failed. The attempt ends, and it is counted towards the number of allowable failed transmissions before the node is prompted to resync with the server.

**Figure B1: Logical flow diagram for node-side mesh network packet handling.** Panel (a.) shows the logical flow for handling a mesh network message transmission attempt. Panel (b.) shows the logical flow for handling a received mesh network message.

In the mesh network model, whenever a node receives a message, it first checks whether it is a broadcast message. If it is not a broadcast message, it is implicitly a unicast ACK message from the server. The node confirms that it is an ACK message and that it is addressed to itself, and if so, records the acknowledgement. If it *is* a broadcast message (either from the server or via a peer), and it is not a message that it has already received, the node will first note the message ID, then process the message contents. If it is addressed to itself, it is implicitly a

broadcast ACK message originating from the server (likely received via a peer). If the node confirms that it is an ACK message with its own address, it records the acknowledgement. If it is not addressed to itself, it could be a data message originating from a peer and addressed to the server; an ACK message originating from the server and addressed to a peer, or a sync message originating from the server and addressed to the entire network. In the first two cases, the node rebroadcasts the message without further processing. In the latter (sync) case, the node first sets its synchronization flag, then rebroadcasts the message to the network.

In a mesh topology network, the server follows the logical flow shown in Fig. B2. First, the server checks to see if the received message is a broadcast or unicast message. If it is unicast, the server returns a unicast ACK. If it is a broadcast message, and if it is not a repeat message, the server broadcasts an ACK message addressed to the originating node.

**Figure B2: Logical flow diagram for server-side mesh network packet handling.** The logical flow for receiving a mesh network message and returning an acknowledgement is shown.

### Data accessibility

SnoTATOS data from the Lincoln Sea case study have been archived at the Arctic Data Center and are available at: doi:10.18739/A2FJ29F6X.

### **Author Contributions**

IAR, DKP, and CMP conceived the study. IAR designed and built the system under study with supervision and feedback from DKP, CMP, and RLH. CMP deployed the sensors. IAR drafted the manuscript. All authors contributed to editing and review. All authors approved the manuscript for submission.

### 499 Competing Interests

The authors declare that they have no conflict of interest.

#### Acknowledgements

- We thank Nathan Kurtz, Patrick Taylor, and the NASA ARCSIX team for collaboration and funding support. We
- thank Patricia Nelsen and Leroy Hessner on the ARCSIX deployment team, pilots Troy McKerrel and Griffin Kelly,
- and the Kenn Borek air team. We thank the Governments of Canada and Greenland and the staff at Eureka,
- Nord/Villum, and CFS Alert Stations for their hospitality and substantial assistance in making the ARCSIX
- deployment possible. We thank David Clemens-Sewall for his contributions to an early version of SnoTATOS, and
- for subsequent conversations and suggestions that greatly improved the system. We thank Marcel Nicolaus for
- similar conversations and critical feedback, and for his help deploying an early version of the system on the
- ARCWATCH expedition. We thank Mario Hoppmann, whose parallel work on distributed snow and ice sensing
- systems helped inform SnoTATOS development, for his feedback on early versions of the system, and for his help
- in the field on the ARCWATCH expedition. We thank Jan Rohde for thought-provoking discussions about sensor
- development and engineering for Arctic science, and for his assistance in the lab and in the field on the
- ARCWATCH expedition. We thank Larson Kaidel, Harvard Brown, Ben Elavgak, and UIC Science for their
- support in deploying a prototype version of SnoTATOS in Utqiagvik in November 2022.

### 515 Financial Support

- IAR received support from the National Defense Science and Engineering Graduate Fellowship. IAR and DKP
- received financial support from NSF OPP-2034919 and NOAA projects 1305M322PNRMT0632 and
- NA24OARX431G0018. CMP received financial support from NSF OPP-2034919, NASA-ARCSIX under IAA-
- 80GSFC23TA046, and the ERDC-CRREL ISOPS program.

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
