# Peer review of "A low-cost, autonomous system for distributed snow depth"

_EGUsphere, 2025_

## Author Response (AR1)

**Response to reviewer #1:**

Thank you very much for your insightful, thoughtful, and encouraging feedback. We particularly appreciate the suggestion from both you and reviewer #2 to highlight the standalone server option, and we have made several modifications to effect this change. Our responses to your specific comments are included below, shown in italics.

- I think that the introduction is interesting and well written. A couple of possible points to strengthen the claims for the value of these measurements. i) These measurements are very useful together with temperature profile measurements, since, as the authors point, the snow is an effective thermal isolation, so that it plays a large role in the temperature profiles obtained. The discussion on this aspect could be slightly extended. ii) there can be significant levels of snow accumulation, which is also adding a "perturbation" to some other measurements. For example, a 1m, or 1.5m, or 2m temperature and / or wind measurement is not any longer done at the same initial height if either 30cm of snow fall, or 30 cm of snow melt, or snow + ice melt takes place. Therefore, having good data on this aspect is also helpful for the accuracy of other kinds of measurements.

*Thank you for these useful recommendations. We have added both points to the introduction as suggested, on lines 44-47 and 52-56.*

- The discussion about the cost savings and methodology are well presented. A couple of additions on this aspect that the authors can consider. Now, the SIMB3 buoy is the key expensive component and may be a major cost driver. Note that it is possible to build also low cost "main instruments", and that using for example iridium modems has become much easier: see, for example, the developments that are happening in the "small buoys" field in several groups around the world: e.g., https://doi.org/10.1016/j.coldregions.2019.102955 , https://doi.org/10.3390/geosciences12030110 , https://doi.org/10.1080/21664250.2023.2249243 , https://doi.org/10.1080/21664250.2023.2283325 , as also reviewed in e.g. https://doi.org/10.48550/arXiv.2410.07813 , and possibly additional similar developments that I may not be aware of. I know (from my own work) that it is easy to interface such solutions with a LoRa modem and / or microcontroller. This means that a further large cost saving could be obtained by also reworking the SIMB3 solution, substituting it by a low cost buoy similar to the ones discussed above. Given the cost

of the SIMB3 buoy, this may allow to further drastically reduce the total cost of the system, and allow true larger scale deployment.

*Thank you for this suggestion and for collecting examples of low-cost "main instruments". We recently completed the development of a completely standalone server that we plan to use as the default option for SnoTATOS deployments in the future. The standalone server is approximately the same cost as a single node (~500 USD), which, as you mention, greatly reduces the cost of a single SnoTATOS network. We have shifted the focus of the paper slightly in order to highlight the standalone option, and we also discuss the characteristics of the standalone server in more detail on lines 220-224 and 294-308. We have updated Figures 1 and 5 to reflect these changes. We are happy to receive any further recommendations or comments regarding these changes.*

- The authors use a MCU (AVR family) + a LoRa modem. Naturally, this is perfectly fine. But I would like to recommend to the authors to consider solutions such as the STM32WL family of MCUs for their future work, that combine the MCU (actually, a significanlty more recent, more power effective, and more powerful one than the "old" 8bit AVR one used here) and the LoRa modem, on a single chip. Having worked both with solutions involving MCU + Lora modem, and solutions involving the integrated STL32WL chip, I can warmly recommend the second one - it is just overall more robust, lower cost, and better to work with; in addition, the stm32duino library support is excellent. But naturally, this is only a minor technical recommendation that the authors may or may not follow in their future design - this is just a minor tips from my experience working with similar solutions.

*Thank you for this suggestion! We will explore the STM32WL family for future SnoTATOS iterations.*

- The choice of the power solution (NiMh batteries) is interesting and a bit surprising to me. Again, "if it works it works", and I do not ask the authors to do any additional work, so this is not a criticism, just sharing my experience. One usually gets significantly better energy density and performance in the cold using primary, non rechargeable Li batteries, such as (there are other suppliers too selling similar batteries) Saft LSH20 or Saft LS33600 batteries (these are also available in different form factors and capacities), depending on the peak power requirement. In the case when no solar panel is attached to the small modules, and they should survive low temperatures (both

of which seem to be the case here), this can be an interesting solution to consider in the future.

*Thank you for this note. We agree that Li batteries are generally the better choice for power density and cold-hardiness, however, their HAZMAT designation makes them difficult to transport (aircraft transport of Li batteries is typically prohibited, and ground and marine transport are heavily regulated). We chose to use NiMH batteries to avoid these logistical barriers. Given the very low power demand of the system, the performance difference between the two battery chemistries was not a primary consideration, and using Li batteries wouldn't significantly impact the system size, weight, or performance.*

- It would be useful to gather all the power consumption data also in a table, and not have these only in the text - this is easier to read for technical people interested in getting a quick overview of the technical facts.

*We have compiled the voltage and power consumption for the nodes, standalone server, and SIMB3 integrated server in Table 2 (Lines 216-217). Please indicate if there is further information that you recommend including in this table, and we will adjust accordingly.*

- The system is described in quite some details, and the main lesson I "take home" is that i) this works reliably, ii) the sensor suggested is probably a good choice (I have been looking at another sensor, but I will investigate this one too in the future). With this information, it will be easy for me to build my own system that is very similar to this one. However, there is no reason to redundantly repeat work across groups. Therefore, my question: it does not seem like you are offering this as an open source project? Would you be able to do so, and to provide mechanical and electrical CAD files and source code on for example a github repository or similar, together with detailed assembly and programming instructions? This would increase a lot the impact of your work and save time and money to the community as a whole.

*Thank you for this suggestion! We plan to provide the materials in an open source repository in the near future, however, we are not immediately prepared to compile and post this repository. This may be possible prior to final publication of this manuscript, in which case we will add a citation for the repository. In the meantime, please feel free to contact us – would be happy to share materials and collaborate on this project.*

**Response to reviewer #2:**

Thank you very much for your close reading of this manuscript and your useful, positive feedback. We appreciate your recommendation to emphasize and discuss the standalone server option in more detail, and have made several changes to reflect this suggestion.

We also note your request to see a more thorough statistical analysis of the representativeness of point sampling snow depth with small sample sizes. We have recently conducted a focused investigation of this topic, and plan to submit it soon as a dedicated manuscript. In the meantime, we have added more detail regarding our recommendations for such work in the context of observing both snow depth/accumulation and surface melt.

Our responses to your specific comments are included in italics below.

1. line 16: 95 and 93% is very exact (and therefore likely fast outdated), I suggest to modify this to something like 'nearly an order of magnitude'

*Thank you for this suggestion. We agree that these are precise values that will likely change as list prices for the various systems change in the future. However, we also feel that phrasing like "an order of magnitude" is ambiguous and does not adequately highlight the significant reduction in cost that this system represents compared to existing platforms. For this reason, we prefer to retain the current phrasing, which cites values that are currently accurate, though they may change in the future.*

2. line 24-25: This same claim is repeated then two more times in the paper. Can you at one of those places (discussion) elaborate this a but better: how?

*Thank you for this recommendation. We have elaborated on this suggestion in the discussion (Lines 469-475).*

3. Line 27-28: here and elsewhere, citations must be ordered chronologically, likely this will be handled by technical editing

*Thank you for your attention to detail! There is some flexibility in the Copernicus/Cryosphere style sheet (https://www.the-cryosphere.net/submission.html) that allows authors to use either chronological, alphabetical, or relevance-based sorting for in-line citations. The in-line citations are currently sorted alphabetically, using the Copernicus formatting option in the Zotero reference manager. However, we do not feel*

*strongly about the citation ordering and are happy to accept any recommendation made during the technical editing stage.*

4. Lines 39-41: This is known from the times of SHEBA (see Matthew Strum et al, 2002, doi:10.1029/2000JC000400). Also recently pointed out by Itkin and Liston in this journal (https://doi.org/10.5194/egusphere-2024-3402)

*Thank you for providing these highly relevant references. We have now cited both in the text (lines 39 and 43).*

5. Lines 55-57: please double check the years of this experiments: N-ICE was from January 2015 to June 2025, MOSAiC was from October 2019 to October 2020.

*Thank you for identifying these errors, we have corrected the dates for both N-ICE and MOSAiC.*

6. Line 58: Here you can also add the Sever data: National Snow and Ice Data Center. (2004). Morphometric Characteristics of Ice and Snow in the Arctic Basin: Aircraft Landing Observations from the Former Soviet Union, 1928-1989. (G02140, Version 1). Romanov, I. P. (Comp.) [Data Set]. Boulder, Colorado USA. National Snow and Ice Data Center. https://doi.org/10.7265/N5B8562T. [describe subset used if applicable]. Date Accessed 03-25-2025.

*Thank you for providing this additional reference. We have included it on Line 70.*

7. Table 1: I would like to see SnoTATOS in two versions here: as extension of SIMB3 and as stand-alone. Also, consider adding IMB buoys developed by bruncin.com. Also, are there no comparable terrestrial systems for snow depth measurements?

*Thank you for this suggestion. We are certainly planning to offer SnoTATOS as a standalone system. We have described this option in more detail, as requested by both reviewers, on lines 220-223 and 284-308. The per station cost will remain the same regardless of whether the system is offered in standalone mode or as a SIMB3 integrated system, and thus the values in this table will not change.*

*We are not aware of comparable terrestrial snow sensing systems. The United States Natural Resource Conservation Service maintains a network of snow observation observation sites (known as SNOTEL sites), however, the instrumentation used at those sites is much more extensive. Campbell Scientific also offers snow depth sensors that can be integrated into their standard dataloggers, and they recently offer an "Internet of*

*Things" edge device module which connects to cellular networks. However, we are not aware of an out-of-the-box distributed snow sensing solution with satellite telemetry.*

8. Lines 125-126: I don't understand the statement 'while conforming to less-than-truckload…' Please, explain how exactly this is relevant. Are you talking about how much weight the ice can bear and how you calculate it relative to long/heavy the nod is? It might e useful to return to this claim in your discussion about the Lincoln Sea case. How it it work there? You did not revisit, but do you have any assumptions based on the data? Did you do any experiments ('in the yard', maybe on a local lake) where you could observe this?

*Thank you for this comment. Less-than-truckload (LTL) shipping refers to transferring freight volumes that are smaller than a standard shipping container/freight truck (e.g., sending individual parcels through DHL or similar). LTL packages are typically required to be less than 2.44 m (8 feet) long.*

9. Figure 3: similar to above comment: The last part of the caption belongs to the main text. If increase of omega with time (melt) all the to the 35 does not matter, this is convincing. How did you measure that?

*Thank you for this suggestion. We have removed the last part of the caption and inserted it into the text on Lines 172-174. We measured the influence of theta on the range reading through a simple experiment: we installed a node with an adjustable adjustable rangefinder in terrestrial snow, and noted the observed value as we varied the sensor look-angle (theta) between 5° and 35°. The range value did not change appreciably over the tested range of theta.*

10. Line 171: Russian Federation not Russia

*Thank you for this note. We have made the suggested change.*

11. Line 226: replace reliable by durable?

*Thank you for this suggestion. We prefer to retain the original since 'reliable' pertains both to the reliability of network communications (i.e., we can rely on successful communications between node and server) as well as the longevity of the system/battery life.*

12. Lines 230-231: Maybe reformulate to be more clear to non-expert reader: theoretical limit for the installation on a perfect flat surface and sensor 1 m over the

ground would yield in 10-20km range. Maybe say something about how high is the stake and how high are normally the pressure ridges.

*Thank you for this recommendation. We have restructured the section for clarity and added the requested information regarding the height of the node and typical ridge height (Lines 270-273)*

12. Line 233: Still, spend a sentence on what is mesh network: I assume nods relay the information…

*We have added a short sentence describing the fundamental concept of the mesh network (Line 274-275).*

13. Line 239: Is this a good place to explain the stand-alone option (or give reference to a different section, appendix)?

*As recommended, we have expanded on the logic flow for the standalone option here.*

14. Line 251: 'hanging'? Is this really how we say it or is it just colloquial expression?

*Thank you for this note. We have changed to "becoming unresponsive" to avoid confusion (Lines 307).*

15. Figure 5: Last sentence of the caption deserved to be in the main text and expanded.

*Thank you for this suggestion. We have added an additional panel to this figure to explain the standalone operation mode, and we have expanded on the description of the standalone system in text. We have edited the figure caption appropriately.*

16. Line 266: 'it pulls the CS line to ground' – I don't understand.

*This means bringing the Chip Select (CS) line down to 0 V (the system ground) to notify the server. We have edited the text to clarify (Line 334-335).*

17. Lines 281-282: Can it be deployed on terrestrial systems?

*Yes, the system can also be deployed in terrestrial environments, which we mention in the Conclusions section (Lines 498-499).*

18. Line 306: 'mean installation conditions': Can you explain how did you collect these data? Was this just one snow depth under each sensor or did you also a floe survey?

*Thank you for this clarifying question. This was just one snow depth/ice thickness at each node. We have modified this line to improve clarity (Line 384).*

19. Figure 8: Why 10-day average?

*The 10-day average preserved the major trends in the data which show significant snow accumulation and surface melt, while smoothing noise that may be due to minor wind redistribution, instrument noise, or other higher frequency processes that occur on 1-2 day timescales. A 10-day average is also a compromise between timescales relevant for observing significant ice growth during the later winter (ice growth was likely on the order of 2-3 cm over 14 days on this relatively thick ice) and timescales relevant for observing ice surface melt.*

20. Lines 348-353: Can you compare this also to the variability at SHEBA, N-ICE and MOSAiC floes?

*Thank you for this suggestion. We have added a brief comparison on lines 434-437 to the SHEBA and MOSAiC floes.*

21. Line 371: Can such difference be justified by looking into weather reanalysis or sea ice deformation (leads) data?

*Thank you for this interesting question. While we believe this is outside of the scope of the current article, the relationship and agreement between in situ snow depth data and reanalysis products is an important topic to pursue in a dedicated investigation. We believe that SnoTATOS is an important tool for pursuing such an investigation, since it enables the collection of precise, distributed snow depth data at comparable scales to reanalysis products.*

22. Line 383-384: Can you give some specific recommendations on the further research and experiment design?

*We have expanded on our recommendations for further research and experimental design on lines 413-419. We note that since submitting this article, we have conducted such a study and will soon submit the work for publication.*

23. Lines 404-408: This is a valuable motivation to develop such system in the first place. Is worth saying this already in the Introduction?

*Thank you for this suggestion. We have added a sentence mentioning SnoTATOS' value for observing terrestrial snowpacks in the Introduction on Lines 100-101.*

---

## Referee Report (RR1)

Raphael et al, 2025: A low-cost, autonomous system for distributed snow depth measurements on sea ice

Second reviewer

I am very pleased with the improvements of the manuscript. Nearly all of my comments were taken into account. The only remaining substantial comment is connected to the comparison of the spatial variability in snow depth to previous studies. This could be done for the initial snow depth conditions and for the development during the accumulation phase. This can be very brief (like it was done for the melt period). Otherwise, I only suggest some minor or technical improvements of text and figures.

(the line numbers refer to the tracked-changes document)

Title: consider including the name of the system into the title. This will be helpful, once your paper is published and it will appear in the reference lists of other publications.

Abstract
lines 20-24: I recommend to write first about the winter and then about the melt period. Potentially, a separate sentence could be added about the accumulation period (if more analysis can be done first). Also, I recommend an explicit clarification about the snow depth variability for data. For example:
'While initial snow depth varied by up to 42% within each network, a comparison of mean initial snow depth between networks showed a maximum difference of only 26%. Similarly, whereas surface melt varied within each network by up to 38%, mean surface melt between networks varied by only up to 9%. This indicates that floe-scale measurements made using SnoTATOS provide a *valuable snow depth variability information and are therefore* more representative data for regional intercomparisons than existing single station systems.

Line 33: conflicts – plural as there is more than one.

Line 45: provide some references for temperature profiles in style 'e.g.'. Consider using references that you already use elsewhere in this manuscript

Line 54: 'similar' or rather 'opposite'

Lines 67-70: maybe begin with the Russian datasets first, so that the data is listed in chronological order

Line 100: is or will be?

Line 170: 'depth stop' is not mentioned elsewhere. I see it was now added to Figure 3. You write somewhere how the change of elevation of the sensor is important and how it may change over the season… Consider writing bit more how this depth stop is useful or not. This is optional.

Line 370 and Figure 6: Please list all network names when you mention them first. Then it will be easier to follow which ones have failed etc. Also, the names in the text and names on Figure 6 are not identical. Please, change the legend on the Figure.

Line 388 (and similarly 402-405, 408, 424-425,...): here you call '2024P', just 'P'. Please be consistent with naming.

Figure 7: Please reduce the time axis on the plot so that times with no data are not shown (29 April-5 January?). Then the plots containing data will be bigger. Also move the 'result description' text from the caption into the main text.

Table 3: 'which snow has filled' to 'snow-filled' for brevity?

Line 465 (and text before): you now added the information about the melt variability comparison to SHEBA and MOSAiC. Please do the same for the snow depth variability during the accumulation period.

Line 490: four nodes were still reporting in January 2025, when you first submitted this manuscript. Would it be worthwhile to update the plots with the complete information now? Surely there is no more data coming in now.